# Gene Expression Profiling as a Potential Tool for Precision Oncology in Non-Small Cell Lung Cancer

**DOI:** 10.3390/cancers13194734

**Published:** 2021-09-22

**Authors:** Sara Hijazo-Pechero, Ania Alay, Raúl Marín, Noelia Vilariño, Cristina Muñoz-Pinedo, Alberto Villanueva, David Santamaría, Ernest Nadal, Xavier Solé

**Affiliations:** 1Unit of Bioinformatics for Precision Oncology, Catalan Institute of Oncology (ICO), L’Hospitalet de Llobregat, 08908 Barcelona, Spain; shijazo@idibell.cat (S.H.-P.); aalay@idibell.cat (A.A.); rmarinm@idibell.cat (R.M.); 2Preclinical and Experimental Research in Thoracic Tumors (PrETT), Molecular Mechanisms and Experimental Therapy in Oncology Program (Oncobell), Bellvitge Biomedical Research Institute (IDIBELL), L’Hospitalet de Llobregat, 08908 Barcelona, Spain; nvilarino@iconcologia.net (N.V.); cmunoz@idibell.cat (C.M.-P.); 3Thoracic Oncology Unit, Department of Medical Oncology, Catalan Institute of Oncology (ICO), L’Hospitalet de Llobregat, 08908 Barcelona, Spain; 4Neuro-Oncology Unit, Hospital Universitari de Bellvitge-ICO L’Hospitalet (IDIBELL), 08908 Barcelona, Spain; 5Program Against Cancer Therapeutic Resistance (ProCURE), Catalan Institute of Oncology (ICO), Bellvitge Biomedical Research Institute (IDIBELL), L’Hospitalet de Llobregat, 08908 Barcelona, Spain; avillanueva@idibell.cat; 6INSERM U1218, ACTION Laboratory, Institut Européen de Chimie et Biologie (IECB), Université de Bordeaux, F-33607 Pessac, France; d.santamaria@iecb.u-bordeaux.fr; 7CIBER (Consorcio de Investigación Biomédica en Red) Epidemiologia y Salud Pública (CIBERESP), 28029 Madrid, Spain

**Keywords:** non-small cell lung cancer, gene expression profiling, transcriptional subtypes, prognosis, targeted therapy, immunotherapy

## Abstract

**Simple Summary:**

It is already known that DNA alterations do not fully recapitulate the complex nature of a tumor or its potential interaction with specific treatments. Therefore, in order to establish more precise and effective therapeutic approaches for non-small cell lung cancer, tumors will have to be characterized in a more accurate and comprehensive way. In this regard, transcription profiling has already demonstrated its utility in further stratifying patients in a much more refined way than genomic alterations. Examples of this include the definition of intrinsic subtypes in colorectal cancer, breast, or non-small cell lung cancer tumors based on their expression patterns. Moreover, the characterization of the activity levels of the pathways involved in tumor progression and development is bound to better predict the specific response to a certain therapy than isolated biomarkers such as specific DNA alterations or the expression of single genes. This is especially relevant in the context of patients not harboring targetable alterations or those developing resistance after treatment.

**Abstract:**

Recent technological advances and the application of high-throughput mutation and transcriptome analyses have improved our understanding of cancer diseases, including non-small cell lung cancer. For instance, genomic profiling has allowed the identification of mutational events which can be treated with specific agents. However, detection of DNA alterations does not fully recapitulate the complexity of the disease and it does not allow selection of patients that benefit from chemo- or immunotherapy. In this context, transcriptional profiling has emerged as a promising tool for patient stratification and treatment guidance. For instance, transcriptional profiling has proven to be especially useful in the context of acquired resistance to targeted therapies and patients lacking targetable genomic alterations. Moreover, the comprehensive characterization of the expression level of the different pathways and genes involved in tumor progression is likely to better predict clinical benefit from different treatments than single biomarkers such as PD-L1 or tumor mutational burden in the case of immunotherapy. However, intrinsic technical and analytical limitations have hindered the use of these expression signatures in the clinical setting. In this review, we will focus on the data reported on molecular classification of non-small cell lung cancer and discuss the potential of transcriptional profiling as a predictor of survival and as a patient stratification tool to further personalize treatments.

## 1. Introduction

Lung cancer ranks first in the number of cancer-related deaths worldwide [1]. Lung cancer patients’ five-year survival rates are strongly related to the tumor stage at the time of diagnosis, ranging from 59% for local disease to 31.7% for regional disseminated disease, and down to 5.8% for distant disseminated disease. At the time of diagnosis, 79% of lung cancer patients show regional or distant dissemination, which subsequently has a dramatic impact on life expectancy [2]. 

The 2015 World Health Organization (WHO) Classification of Lung Tumors divides lung cancer into two main histological types: epithelial and neuroendocrine tumors. Epithelial tumors, often also called non-small cell lung cancer (NSCLC), account for 80–85% of all cancer diagnoses and are subsequently subdivided into two major subtypes: non-squamous cell carcinomas, further divided in lung adenocarcinoma (LUAD) and large cell carcinoma, and squamous cell carcinomas (LUSC) [3]. The TNM staging criteria has been widely used for the management of patients with NSCLC, both as a prognostic factor and as a tool for treatment decision making. Nevertheless, the current system based almost solely on the tumor histology and morphology is unable to completely explain the complexity of this pathology. For instance, similar tumors in terms of histology or pathological stage do not always follow the same clinical behavior or display equal responses to the same treatment. In fact, between 30 to 55% of early-stage NSCLC patients relapse and die of the disease despite complete resection with clear resection margins [4]. 

Technological advances within the last decades have led to a significant improvement in our understanding of tumor biology through the implementation of large-scale genomic and transcriptomic analyses. In this way, whole-transcriptome gene expression profiling, coupled with bioinformatics analyses, has been used to identify distinct NSCLC molecular subtypes [5]. However, the clinical relevance of those classifications has been questioned and surpassed by the identification of actionable drivers, which had a major impact on the molecular classification and treatment landscape of lung cancer, especially in lung adenocarcinoma.

In this review, we attempt to analyze the current state of transcription-based classifications in the context of NSCLC and lay the groundwork for future studies aiming to define a transcriptional classification of NSCLC with clinical value, both as a predictor of survival and as a patient stratification tool to further personalize treatments. This is particularly relevant since a significant proportion of patients lack actionable targets and are currently being treated with chemotherapy alone or in combination with immunotherapy. 

## 2. NSCLC Intrinsic Molecular Subtypes

Gene expression is intimately linked to cellular phenotype and tumor behavior. Messenger RNA (mRNA) expression has been extensively used to distinguish biologically homogeneous subtypes of a disease. In lung cancer, seminal studies on transcriptional profiling included different morphological types (i.e., lung LUAD, lung LUSC, large cell lung cancer, and small cell lung cancer) in order to classify them based on their differential gene expression [6,7]. These studies demonstrated that gene expression profiling can recapitulate the morphologic classification of NSCLC and that, unlike other morphological subtypes, lung LUAD could be further subclassified into several transcriptional groups. However, the lower prevalence of the other histologies, compared to LUAD, could have hindered the identification of molecularly different subtypes within these morphological entities. As a consequence, subsequent studies generally focused on one histological type, especially LUAD or LUSC, due to their higher abundance (Table 1, Figure 1, Appendix A). 

### 2.1. Transcriptional Subtypes in Lung LUAD

Following the evidence of LUAD’s higher molecular heterogeneity, a number of studies developed between 2001 and 2016 sought to generate a classification of lung LUAD based on gene-expression profiling [6,7,8,9,11,13,15,17,18,20,21,22,24,25] (Figure 1A, Appendix A). All these studies converged in the identification of a subset of predominantly well-differentiated LUAD tumors with higher expression of pneumocyte markers and associated with better prognosis. Additionally, another group of poorly differentiated tumors with higher expression of cell proliferation-related genes and associated with poor survival outcomes was also commonly identified. Nevertheless, the variation in the number of additionally identified subgroups, unique to each study, has prevented the definition of a consensus classification (Figure 1). 

To date, the most accepted classification was described back in 2006 [11]. Hayes et al. described three consensus transcriptional subtypes (e.g., bronchioid, squamoid, and magnoid), combining publicly available gene expression datasets [6,7,8]. Bronchioid tumors were more frequent among non-smoking females and showed better survival outcomes in an early-stage disease context compared to the other groups. Interestingly, this trend was inverted in late-stage tumors, where magnoid and squamoid subtypes showed better survival rates. Besides, these subtypes also displayed differentially expressed transcriptional programs involved in different biological processes. In this way, bronchioid tumors showed higher pneumocyte type II and cisplatin resistance-related gene expression, whereas squamoid and magnoid tumors were found to be associated with overexpression of immune system and inflammatory-related genes, respectively. Six years later, in 2012, these groups were revisited for associations with different genomic alterations, known to be important for the biology of NSCLC [20]. *EGFR* alterations were mostly present on bronchioid tumors, whereas *KRAS* and *TP53* were more common within the magnoid subtype. In addition, magnoid tumors exhibited the highest chromosomal instability, copy number alterations, DNA hypermethylation, and genome-wide mutation rates. 

Hayes et al.’s and Wilkerson et al.’s [11,20] classification was then applied to The Cancer Genome Atlas (TCGA) LUAD dataset comprising 230 lung LUAD [24]. However, transcriptional subtypes were renamed according to additional morphological, histological, and molecular characteristics: terminal respiratory unit (TRU, bronchioid), proximal inflammatory (PI, squamoid), and proximal proliferative (PP, magnoid). As reported in previous studies [11,20], these subtypes were associated with specific DNA alterations. The PP subtype showed the highest *KRAS* mutation and STK11 inactivation frequency, while the PI subtype was enriched for *NF-1* and *TP53* co-mutation. On the other hand, activating *EGFR* mutations or ALK rearrangements were more prevalent within the TRU subtype. After all, this classification did not contribute much to the previous one but served as an independent validation of Hayes et al.’s and Wilkerson et al.’s intrinsic subtypes in a larger set of tumors. 

Over the years, the TCGA project has been incorporating new tumor cases, which means a doubling increase in the number of lung LUAD cases with available molecular data. This opened the opportunity for further research on the establishment of accurate and clinically relevant tumor molecular subtypes. Hu et al. combined k-means clustering, *t*-test, sensitivity analyses, self-organizing map (SOM) neural networks, and hierarchical clustering methods and reported a new classification into four intrinsic lung LUAD subtypes [27], named 1 to 4. Again, these subtypes showed association with different biological processes. For instance, groups 1 and 2 overexpressed immune-related signaling pathways, whereas subtypes 3 and 4 showed higher activation of cell proliferation and extracellular matrix organization pathways, respectively. Moreover, an association was also reported between gene expression subtypes and specific mutations. In this way, *EGFR* alteration was enriched in subtype 4, while *TP53* mutations tended to be more frequent in subtypes 1 and 2. In another study, Chen et al. reported a novel classification consisting of six lung LUAD subtypes (e.g., AD1-AD4, AD5a, AD5b), integrating gene expression, DNA methylation, copy number alterations, and protein expression data [26]. In agreement with previous classifications, LUAD subtypes associated with low differentiation levels showed relatively worse prognosis. Moreover, an association between these subtypes and specific transcriptional programs and biological processes was also reported. 

### 2.2. Transcriptional Subtypes in Lung LUSC

Although most of the previously mentioned studies focused on lung LUAD, some studies also reported intrinsic transcriptional subtypes for LUSC (Figure 1B, Appendix A). Between 2005 and 2007, the first attempts at defining subgroups in LUSC by Inamura et al. [10], Larsen et al. [14], and Raponi et al. [12] identified two major clusters of tumors associated with different survival outcomes and differentiation grades. 

In a more comprehensive study published in 2010, Wilkerson et al. proposed four intrinsic LUSC subtypes: primitive, secretory, basal, and classical [16]. These groups were significantly associated with survival, tumor grade and distinct biological processes. Primitive LUSC showed overexpression of proliferation-related genes and the worst overall survival among subtypes. Classical tumors overexpressed genes involved in xenobiotic metabolism. Secretory LUSC showed higher expression of genes involved in immune response and pneumocyte type II markers. Basal subtype presented higher expression of genes involved in cell adhesion and basement membrane functions. In terms of prognosis, basal, secretory, and classical tumors showed similar survival outcomes, better in general compared to primitive subtypes. These results were further discussed by Brambilla et al., who identified an additional basaloid-like subtype, found to mostly agree with Wilkerson et al.’s primitive subtype but far more associated with a poor prognosis phenotype [23].

In 2012, Wilkerson et al.’s LUSC transcriptional subtypes’ were tested in the TCGA LUSC dataset, along with other available molecular data, to classify and characterize 178 lung LUSC samples [19]. Again, correlations were observed between the different transcriptional subtypes and genomic alterations in copy number, DNA mutations, and methylation. The classical subtype showed higher *KEAP1*, *NFE2L2*, and *PTEN* alterations frequency, as well as pronounced hypermethylation and chromosomal instability. By contrast, primitive tumors more commonly exhibited *RB1* and *PTEN* alterations, while basal tumors showed *NF1* alterations. Interestingly, *CDKN2A* deletions, which are a common event in LUSC, were not associated with any subtype.

In 2017, Chen et al. took advantage of the incorporation of new LUSC tumors within the TCGA resource and delved into the study of LUSC subtypes combining omics layers other than transcriptomics [26]. Three LUSC subtypes associated with transcriptional targets of SOX2 or TP63, as well as cancer-testis expression, were identified. 

### 2.3. Current Clinical Applicability of NSCLC Gene Expression Signatures

Overall, different classifications have been proposed for both lung LUAD and LUSC but have never been adopted within the clinical practice. Gene expression signatures derived from these studies, generally conducted on a limited number of patients, sometimes included large sets of genes that passed the significance threshold. This often makes it difficult to select the most relevant genes, leading to less reproducible results or identification of the relevant biological processes that these genes represent [28]. In fact, not surprisingly, little overlap was found between the gene expression signatures derived from some of these studies (Appendix A). In addition, the fact that most of them have never been accurately validated in prospective studies to prove their clinical benefit has prevented their introduction into the clinical setting. Furthermore, the increased complexity in their interpretation favors the use of genomic sequencing, which allows identifying discrete actionable drivers that have been shown to predict clinical benefit from targeted therapies [29]. 

In this regard, in addition to the signatures derived from lung cancer molecular subtypes studies, several groups designed studies whose main objective was to identify prognostic signatures, some of them also able to predict clinical benefit from specific treatments, using gene expression profiling in surgically resected tumor samples. In 2017, Tang et al. evaluated 42 of these signatures for their prognostic potential via a meta-analysis on almost 2000 early-stage NSCLC patients collected from 15 studies [30]. Although more recent studies have been published between 2016 and 2021 (Appendix A), to our knowledge, this is the most recent review that comprehensively evaluates prognostic gene expression signatures in NSCLC [30]. Unlike a previously published study reporting a lack of utility of some of these signatures on top of clinical risk factors [31], Tang et al. identified 25 signatures able to predict survival outcome adjusting for clinical risk factors. Moreover, 18 of them significantly outperformed random signatures. These differences in the findings of these two studies mainly reside in the different nature of their design. Subramanian and Simon et al. [31] performed a critical review of the aim, methodology, results, and derived conclusions of the evaluated studies, whereas Tang et al. [30] systematically compared the prognostic performances of the published signatures.

Although distant in time, both studies agreed on the limitations that can be encountered when evaluating and proposing prognostic signatures for their clinical use. Some of these caveats, which also apply to recent studies (Appendix A), include low reproducibility of the analysis from where the signatures come from, inconsistent patient’s eligibility criteria, lack of prospective validation, and the fact that most gene expression signatures are based on microarray platforms which require good quality fresh frozen (FF) tissue samples. This requirement prevents the implementation of prognostic signatures in the routine clinical practice, where FF tissue samples availability is often scarce [32]. In this regard, only a few attempts to develop mRNA signatures adapted for use in formalin-fixed paraffin-embedded tissue (FFPE) samples have been made [33,34,35,36]. However, prospective validations of these signatures are needed before they can be used in the clinic and potentially solve tissue availability limitations. 

## 3. Gene Expression Profiling in the Context of Targeted Therapies in NSCLC

NSCLC is not a single entity but is, in fact, multiple pathologies, each with unique molecular features, which we are only beginning to understand [37,38]. Even though the distinction between distinct histological subtypes is clinically relevant and has an impact on therapeutic decisions, the current management of NSCLC requires tumors to be screened for specific genomic alterations that predict survival benefit and sensitivity to targeted therapies [39]. However, there are patients lacking tractable genomic alterations, and patients with oncogenic drivers may respond differently to targeted therapies for reasons that remain unclear, and all patients will eventually develop treatment resistance. Therefore, the implementation of new methodologies beyond genomic testing, such as those based on gene expression, will be crucial to delivering more precise and effective treatments to NSCLC patients (Figure 2).

### 3.1. Gene Expression Profiling as a Tool for the Stratification of Driver-Positive Patients

Current biomedical research is focused on developing treatments that specifically target abnormally activated regulatory pathways of cancer cells, such as signal transduction, cell cycle, DNA repair, metabolism, and apoptosis resistance. Significant advances in understanding the genomic landscape of lung cancer through comprehensive genomic profiling have allowed physicians to tailor treatment options in many cancer types, including lung cancer [40,41]. 

Around 38% of patients with NSCLC present an oncogenic driver mutation, which enables the use of targeted treatment options that improve survival outcomes and reduce toxicity compared to standard chemotherapy [42,43,44]. EGFR mutation and EML4-ALK rearrangement are paradigmatic examples of how biomarker-driven targeted therapy has shifted the treatment of patients with NSCLC [44]. Unfortunately, not all patients respond equally to these therapeutic approaches. Recently, intratumor heterogeneity, concurrent genomic alterations, and pre- or post-treatment heterogeneity regarding targetable oncogenic mutations can partly explain these differences in response and duration of clinical benefit to targeted therapies [45]. 

Gene expression analysis could become a powerful tool to further stratify patients for targeted therapies and to predict potential differences in response to treatment among patients harboring the same genomic alteration. In this regard, there are only a few studies concerning the comparison of driver alterations and defined transcriptional subtypes in the context of NSCLC [13,15,19,20,24,26,27,46,47,48] (Table 2). Although transcriptional-based NSCLC molecular subtypes derived from these studies were enriched for key genomic drivers such as *EGFR* and *KRAS* mutations, the transcriptional signatures allowed further refinement of disease subclassification. In this regard, only two of these studies found specific transcriptional profiling patterns for patients harboring EGFR or ALK alterations, indicating that the presence of these alterations may confer a very specific gene expression pattern [46,48]. On the other hand, the majority of studies found patients with these alterations in all the identified subtypes, although there was almost always one that was enriched in a certain alteration (i.e., EGFR) [13,15,20,24,25,26,27,47]. In this context, *EGFR* mutated tumors are still considered to be a relatively homogeneous entity for therapeutic decisions in NSCLC, despite drug responses and time to progression are relatively variable among patients. Moreover, gene expression profiles of tumors harboring mutations in other key genes such as *KRAS* or *TP53* showed significant heterogeneity [8,13,15,19,20,24,25,26,27,47,49]. In particular, *KRAS* is the most common gain-of-function mutation in NSCLC, accounting for approximately 30% of lung AC in Western countries and about 10% in Asian countries [50]. Recent advances in drug development yielded novel promising agents like sotorasib and adagrasib, which target KRAS-G12C and will soon change the therapeutic landscape of *KRAS* mutated tumors [51]. Given the high heterogeneity in terms of genomic and transcriptomic profiles of KRAS mutant NSCLC, it is reasonable to expect wide-ranging outcomes of those targeted agents. 

In summary, this evidence supports the fact that DNA alterations do not fully recapitulate the complex nature of lung tumors. Therefore, further stratification of these patients based on their transcriptional profiling might improve the selection of individuals that would benefit from existing targeted therapies. Moreover, advances in this direction could lead to the development of novel combinations with standard care to target alternative signaling cascades in patients suspected to be non-responders or the anticipation of resistance mechanisms and the design of combination treatments upfront. 

### 3.2. NSCLC Tumors Lacking a Tractable Oncogenic Driver

Targeted DNA sequencing has recently been demonstrated to be a comprehensive tool to detect multiple types of oncogenic alterations, including relevant oncogenic kinase fusions in NSCLC (i.e., ROS1, ALK, RET). Currently available DNA-based panels, such as FDA-cleared MSK-IMPACT large panel, are designed to detect rearrangements via tiling of the appropriate introns known to likely harbor the genomic alteration [52]. However, this approach has shown some limitations, including difficulty to tile very long introns, the existence of unmappable repetitive elements, and the possibility of the fusion event occurring in an alternative intron not included in the panel design [53]. This may cause patients to be considered as driver-negative, although they may indeed hold a targetable fusion. In this regard, RNA-Seq seems to represent a more direct approach for fusions detection in those cases, as introns are removed during splicing events. Following this reasoning, Benayed et al. carried out a retrospective sequencing analysis using an RNA-Seq based fusion panel on LUAD samples reported to be driver-negative by a DNA-based sequencing platform [54,55]. The results of the study showed that 14% (n = 36) of negative patients after targeted DNA-Seq were positive for fusions by targeted RNA-Seq and could benefit from corresponding targeted therapy, thus demonstrating the importance of following DNA-Seq by RNA-Seq for fusion detection. Of note, RNA-Seq used in this study was performed using a panel with a limited number of genes. Therefore, the use of whole transcriptome sequencing will allow the detection of targetable fusions, including some not yet described but potentially relevant, in patients deemed driver negative [56]. 

On the other hand, a significant percentage of NSCLC patients do not harbor any known actionable genomic alterations, especially in LUSC, for whom there are currently no available targeted therapies. For this reason, there is a need to unravel novel vulnerabilities other than genomic oncogenic alterations and to combine new technologies that allow a deeper understanding of cancer biology, such as those centered on gene expression. In this regard, the WINTHER and CoPPO trials evaluated the value of transcriptomics in addition to genomic sequencing to guide treatment decisions in advanced malignancies that have progressed following standard treatments [57,58]. Both studies showed that incorporating transcriptomics increased the number of treated patients compared to assessing only genomic alterations. Indeed, the use of gene expression profiling increased the matching rate among genomic alteration and treatment compared with previous genome-guided therapy studies [57]. 

Overall, these trials showed that transcriptomic profiling beyond classical tumor DNA sequencing is useful to make treatment decisions in patients with solid tumors. Therefore, future research should seriously consider combining genomic and transcriptomic data.

### 3.3. Overcoming Targeted Therapy Resistance Mechanisms

Although the use of targeted therapies has led to clinical benefit in patients with NSCLC, responses to these treatments are rarely complete and inevitably temporary because of the appearance of treatment resistance mechanisms. 

There are two main types of resistance mechanisms to targeted drug therapy: primary resistance or acquired resistance. Additionally, these processes can be further subclassified as on-target and off-target [59]. Regarding on-target resistance, it arises when the target of the drug itself is mutated, leading to response attenuation or even causing no response at all. On the other hand, off-target mechanisms occur through activation of signaling cascades parallel or downstream the drug target and, therefore, bypassing the potential inhibitory effect of the treatment. However, off-target resistance mechanisms may not necessarily be genetic. In this way, alterations in signaling pathways, histological and phenotypic transformations during tumor progression [60,61], epigenetic changes that favor the survival of drug-tolerant tumor cell subpopulations [62], and interactions between the tumor and its microenvironment [63,64], can modify tumor cell sensitivity to the targeted therapy. All these mechanisms may not be captured at the genomic level but can indeed have an impact on the transcriptional profile of the tumor. Moreover, most of the identified genomic mechanisms of resistance tend to converge into recurrent signaling pathways, which promote tumor growth and cell survival. Hence, the identification of transcriptional programs deregulated upon targeted therapy through transcriptional profiling could be a pragmatic approach to capture and tackle drug resistance [59]. 

EGFR-TKIs are widely used against NSCLC with EGFR mutations. Thus, preventing and delaying EGFR-TKI resistance is a critical clinical concern involving the treatment of EGFR-mutant NSCLC [65]. In the past years, preclinical models have been extremely helpful to identify specific mechanisms of drug resistance. For instance, MAPK pathway reactivation upon EGFR and ALK inhibition has been reported to occur at different points in the signaling cascade due to specific mutations [66,67,68,69,70]. More interestingly, this resistance could be reversed and even prevented through combination therapy with MEK and EGFR or ALK TKI. Several pathways have been found upregulated upon resistance, such as PI3K-AKT [71,72], JAK-STAT [73,74], and IGFR1 [75]. However, one of the main limitations of most studies focused on assessing abnormal pathway activation that may be responsible for certain acquired resistances upon treatment, is that they are based on determining the master regulator protein level through experimental molecular biology techniques. Although the identification of these altered signaling cascades is important, these approaches give little information about the mechanisms that led to such alterations, which could, for instance, be deregulating other pathways that may not be the focus of these studies but, that could as well be promoting resistance to treatment and tumor progression. Moreover, deregulation of compensatory pathways cannot be captured with targeted next-generation-sequencing (NGS) panels and, therefore, whole transcriptome analysis of preclinical models or tumor samples obtained at the times of diagnosis and treatment resistance might become essential for the identification of non-canonical mechanisms of resistance that may eventually be translated into novel therapeutic approaches to overcome resistance to targeted therapies. In this regard, few studies have attempted to use whole-transcriptome microarray/RNA-Seq techniques in order to elucidate the mechanisms that lead to EGFR-TKI sensitivity or resistance (Table 3). However, the immediate clinical applicability of these studies is complicated. First, most of these studies were carried out using pharmacogenomics data from large cancer cell lines projects, which has been challenged before due to sensitivity metrics inconsistencies between projects driven by inherent cancer cell lines heterogeneity. Moreover, when cellular models are used, all the information regarding the tumor microenvironment is lost, which is a big concern considering that interactions between tumor cells and the microenvironment could also be determining treatment resistance [76]. Therefore, we believe that the implementation of these signatures or the discovery of new ones for clinical use will depend on the execution of clinical trials with matched gene expression and treatment effect information for NSCLC patients. Moreover, as significant variability has been observed in drug response at the level of individual cells within a heterogeneous cellular population, the use of single-cell RNA-Seq techniques and the development of standardized analysis pipelines is bound to result in the identification of specifically which cells are responsible for treatment failure and might unravel new targets. 

## 4. Gene Expression Profiling in the Context of Immunotherapy in NSCLC

### 4.1. Gene Expression Signatures as Predictors of Immunotherapy Response

Until recently, the standard first-line therapy for patients with metastatic or advanced NSCLC has been doublet platinum chemotherapy [87]. Nowadays, immune checkpoint inhibitor (ICI) therapy against the programmed cell death-1 (*PD-1*) or its ligand *PD-L1* axis, either alone, in the case of patients with *PD-1* expression ≥ 50%, or in combination with chemotherapy, in the case of *PD-1* expression < 50%, has become the standard of care in advanced NSCLC lacking driver oncogenic alterations [88]. However, some patients will progress early to these treatments and the rate of long-term survivors is relatively low. 

Previously established biomarkers for predicting clinical outcomes of immunotherapy in NSCLC, such as PD-L1 expression, tumor mutational burden (TMB) or single genomic alterations in STK11, KEAP1, or PTEN genes, cannot accurately predict clinical benefit in all patients [89]. In this regard, none of these biomarkers is sufficient to predict tumor response or longer survival outcome to immunotherapy or combined immunotherapy plus chemotherapy. 

Gene expression profiling, interrogating several immune-related genes, is becoming an attractive approach to characterize tumor immune microenvironment and predict immunotherapy response, going beyond the measurement of single genes such as PD-L1 (Table 4). In part, this is because immune populations infiltrating the tumor can be detected and characterized thanks to transcriptional profiling. 

A recent study demonstrated that multigene signatures constitute a reliable tool to identify T-cell pro-inflammatory phenotypes across different solid tumors, including NSCLC, likely to respond to PD-(L)1 inhibition therapy outperforming PD-L1 immunohistochemistry in PD-L1 unselected patients [94]. Of note, the 18-gene T cell-inflamed gene expression profile derived from this study was assessed using the Nano String nCounter platform, which is compatible with formalin-fixed paraffin-embedded samples much more suitable than fresh frozen tissue in the clinical setting. Moreover, this signature underwent further validation and is under development as a clinical-grade diagnostic device, currently being used in a set of ongoing ICI therapy trials [98]. Soon after, Hwang et al. evaluated the immune-related gene expression profile of prospectively gathered tumor samples from patients with metastatic NSCLC, who had not received prior treatment, to find putative sensitivity or resistance-related mechanisms to ICI therapy [99]. All patients were treated with anti-PD-1 and were divided into two groups depending on whether they achieved a durable clinical benefit or not. The results showed that gene expression signatures, specifically T-cells and M1 macrophages from MSigDB C7 collection, were able to discriminate between durable responders and non-durable responders. Although a prospective validation is needed, the authors concluded that these gene sets were better predictors than PD-L1 status or TMB. 

Overall, these two studies showed that gene expression profiling constitutes a powerful tool for predicting response to immunotherapy outperforming previously single analyte biomarkers. However, both used multiplexed panels containing a few selected genes, thus limiting the amount of information that can be captured compared to whole transcriptome sequencing. In this regard, we believe that further studies introducing whole transcriptome sequencing, with the subsequent development of standardized single-cell RNA sequencing analysis pipelines to properly analyze the composition of complex mixtures of cells, would be useful not only to distinguish which patients will derive long-term clinical benefit from immunotherapy, but also to characterize the mechanisms of acquired resistance to anti-PD-(L)1 treatment, and to progress towards a more precise immunotherapy. 

### 4.2. Linking Lung Cancer Molecular Subtypes with Immune Phenotype

Given the lack of accurate biomarkers for immune therapy response in NSCLC, gene expression profiling was explored for lung LUAD and LUSC. Faruki et al. [100] assessed the immune landscape of previously defined lung intrinsic subtypes of LUAD (TRU, PP, PI) and LUSC (classical, primitive, secretory, and basal). They measured the abundance of different immune cell populations involved in both adaptative and innate immune responses using publicly available data from 1190 lung LUAD [9,20,24,101] and 761 lung LUSC [12,16,19,102]. They observed differences in the immune cell infiltration across intrinsic subtypes, with decreased immune cell expression in the PP LUAD subtype and high immune cell expression in the secretory and primitive LUSC subtypes. They also evaluated the correlation between immune cell expression and PD-L1 gene expression. In LUSC, immune cell infiltration was not correlated with PD-L1 IHC expression. In this regard, in the phase III clinical trial CheckMate-017, tumor PD-L1 expression did not predict benefit from nivolumab in patients with advanced lung LUSC [103], showing inconsistencies with other studies reporting this positive correlation [104]. In addition to PD-L1, TMB was also reported to be associated with immunotherapy response in NSCLC [105]. However, Faruki et al. did not find an independent association between TMB and immune cells expression either in LUAD or in LUSC subtypes. In this regard, the PI LUAD subtype does show this positive correlation between TMB and immune infiltration, whereas the TRU subtype presents the lowest mutational burden despite quite high immune cells expression. Similar findings were reported for the association of certain genomic alterations with immune system impairment. In this way, STK11 loss/inactivation in LUAD or KEAP1/NFE2L2 pathway alterations in LUSC were differentially enriched between intrinsic subtypes but were consistently associated with lack of immune cells expression. Interestingly, although KEAP1 alterations have been previously reported to be associated with high TMB, this correlation resulted in little clinical benefit for immunotherapy treatment [97]. The authors claim that this should be taken into consideration in future studies, as the proportion of LUAD or LUSC subtypes within the study population could have a major impact on the results regarding the stratification of NSCLC patients for immune-checkpoint treatment based on TMB or genomic alterations. Overall, this work presents distinct immune infiltration patterns associated with intrinsic LUAD and LUSC gene expression-based subtypes. More interestingly, these differences in the tumor microenvironment as well as within tumor biology could potentially predict immunotherapy response and provide for better patient stratification in future drug trials.

## 5. Conclusions

Transcriptomics has not historically had an excessively relevant role in the diagnosis and treatment of NSCLC. Intrinsic technical and analytical difficulties (i.e., variability of the results, costs, tissue sample availability, challenging interpretation of the outcomes), in addition to the weight of genomics in the management of NSCLC have contributed to hinder its use within the clinical practice.

Future clinical development of targeted therapies relies on the integration of all the accumulated knowledge on NSCLC biology. The definition and characterization of molecularly homogeneous NSCLC subtypes is bound to result in the design of effective therapeutic strategies. To this end, it is crucial to move away from characterizing tumors just on the basis of single biomarkers, which are so far unable to fully predict the complexity of the disease and drug response. In this way, we should evolve towards the integration of different technologies that allow us to take into account clonal heterogeneity, the development of resistance mechanisms, and interactions with the tumor microenvironment. In this context, transcriptome profiling has been demonstrated to be a powerful tool for patient stratification and drug selection, especially in the context of acquired drug resistance and patients with advanced NSCLC lacking actionable drivers. The relevance of transcriptional profiling will likely increase with the progress of immune therapy and the need for predictive markers. Moreover, the recognition of signaling pathways altered upon tumor progression through gene expression profiling has conferred new opportunities for drug development and has increased the number of patients that can be matched to a treatment. In addition, the identification of associations between the altered pathways and the presence of specific mutational patterns responsible for acquired resistance and tumor progression might unravel new therapeutic targets. In this context, there have recently been important developments in the field of transcriptomics in order to increase resolution and specificity. For instance, single-cell RNA-Seq allows the determination of the gene expression profiles of individual cells. In this way, it provides information about tumor heterogeneity and cellular interactions. Moreover, it also provides insights into which cells might be responsible for resistance development and through which transcriptional processes, exposing new targets for drug development.

Nevertheless, the implementation of whole-transcriptome interrogation techniques into NSCLC management frameworks will mainly depend on prospective validation studies of the identified signatures to assess whether they are able to improve treatment decisions and recapitulate the heterogeneity of the disease beyond currently available strategies.

## Figures and Tables

**Figure 1 cancers-13-04734-f001:**
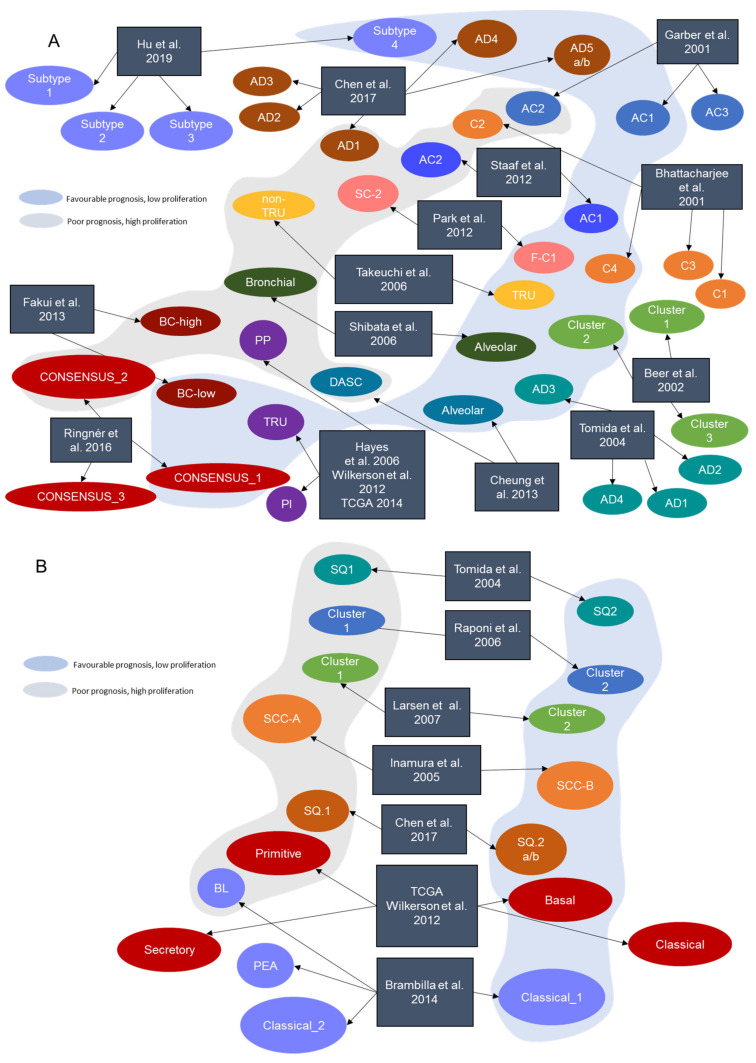
Intrinsic NSCLC transcriptional subtypes. Some of the NSCLC transcriptional subtypes described by different research groups are clearly comparable, based on prognostic and molecular characteristics (same color shadows). (**A**) Transcriptional subtypes described in LUAD. (**B**) Transcriptional subtypes described in LUSC. In both histologies, the blue shadow represents a set of tumors with better prognosis and low proliferation-related pathways activity whereas the grey shadow represents a subset of tumors with worse prognosis and a proliferative and more aggressive behavior.

**Figure 2 cancers-13-04734-f002:**
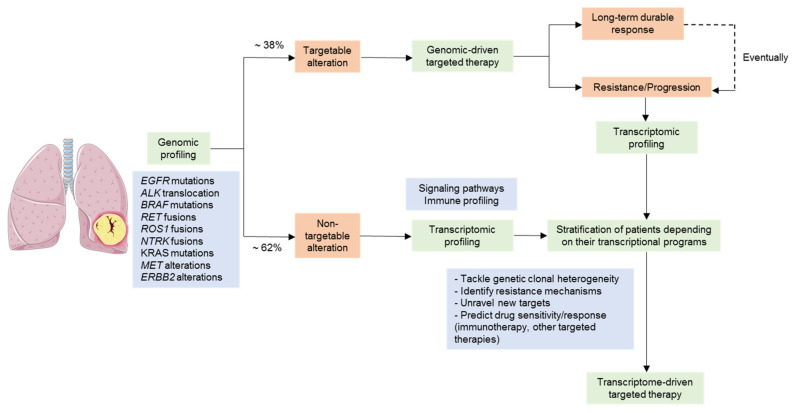
Transcriptional profiling as a tool for guiding precision medicine treatments in NSCLC. Flowchart representing key situations where gene expression profiling can add to genomic profiling in the management of NSCLC treatment. Molecular testing to detect targetable alterations is required in all non-squamous NSCLC to detect genomic alterations that cause an oncogenic addiction and for which there is an available targeted therapy (i.e., *EGFR*). Although targeted therapies contributed to a significant improvement of the survival time in lung cancer patients harboring genomic alterations, patients will progress within several months due to the emergence of resistance mechanisms. More importantly, a significant proportion of patients do not harbor any actionable driver alteration. To incorporate new technologies, such as those centered on gene expression, might allow unraveling novel vulnerabilities other than genomic oncogenic alterations.

**Table 1 cancers-13-04734-t001:** Non-small cell lung cancer gene expression subtypes signatures.

Signature	Year	Journal	Reference	Tissue of Origin	Number of Genes	Identified Subtypes	Technology
Bhattacharjee et al.	2001	*Proc. Natl. Acad. Sci. USA*	[7]	LUAD	100	C1, C2, C3, C4	Microarray
Garber et al.	2001	*Proc. Natl. Acad. Sci. USA*	[6]	LUAD	146	AC1, AC2, AC3	Microarray
Beer et al.	2002	*Nat. Med.*	[8]	LUAD	4966	Cluster 1, Cluster 2, Cluster 3	Microarray
Tomida et al.	2004	*Oncogene*	[9]	LUAD, LUSC	829	AD1, AD2, AD3, AD4, SQ1, SQ2	Microarray
Inamura et al.	2005	*Oncogene*	[10]	LUSC	432	SCC-A, SCC-B	Microarray
Hayes et al.	2006	*J. Clin. Onc.*	[11]	LUAD	2553	Bronchioid, Squamoid, Magnoid	Microarray
Raponi et al.	2006	*Cancer Res.*	[12]	LUSC	11,101	Cluster 1, Cluster 2	Microarray
Takeuchi et al.	2006	*J. Clin. Onc.*	[13]	LUAD	293	TRU, non-TRU	Microarray
Larsen et al.	2007	*Carcinogenesis*	[14]	LUSC	6748	Cluster 1, Cluster 2	Microarray
Shibata et al.	2007	*Cancer Sci.*	[15]	LUAD	78	Alveolar, Bronchiolar	Microarray
Wilkerson et al.	2010	*Clin. Cancer Res. Off. J. Am. Assoc Cancer Res*	[16]	LUSC	208	Primitive, Basal, Secretory, Classical	Microarray
Park et al.	2012	*PloS ONE*	[17]	LUAD	191	S_C1, F_C2	Microarray
Staaf et al.	2012	*BMC Med. Genomics*	[18]	LUAD	176	AC1, AC2	Microarray
TCGA-LUSC	2012	*Nature*	[19]	LUSC	208	Primitive, Basal, Secretory, Classical	RNA-Seq
Wilkerson et al.	2012	*PloS ONE*	[20]	LUAD	506	Bronchioid, Squamoid, Magnoid	Microarray
Cheung et al.	2013	*Cancer Cell*	[21]	LUAD	249	Alveolar, Distal airway stem cell-like (DASC)	Microarray
Fukui et al.	2013	*Eur. Respir. J.*	[22]	LUAD	1829	BC-low, BC-high	Microarray
Brambilla et al.	2014	*Clin. Cancer Res. Off. J. Am. Assoc. Cancer Res*	[23]	LUSC	139	Classical_1, Classical_2, PEA, BL	Microarray
TCGA-LUAD	2014	*Nature*	[24]	LUAD	506	TRU, PI, PP	RNA-Seq
Ringnér et al.	2016	*Oncotarget*	[25]	LUAD	Consensus classification	CONSENSUS_1, CONSENSUS_2, CONSENSUS_3	Consensus
Chen et al.	2017	*Oncogene*	[26]	LUAD, LUSC	700	AD.1, AD.2, AD.3, AD.4, AD.5a, AD.5b, SQ.1, SQ.2a, SQ.2b	RNA-Seq + Other omics
Hu et al.	2019	*Trans. Comput. Biol. Bioinform.*	[27]	LUAD	30	AD.1, AD.2, AD.3, AD.4, AD.5a, AD.5b	RNA-Seq

**Table 2 cancers-13-04734-t002:** Studies concerning the association between genomic alterations and specific transcriptional profiles. * Studies with * had whole-exome sequencing data, and as a wide range of mutations were evaluated with this technology, only the most relevant ones were reported on this table.

Signature	Histology	Evaluated Genomic Alterations	Associated Subtype
Takeuchi et al. [13] 2006	LUAD	*EGFR*	TRU-type
		*KRAS*	-
		*TP53*	-
Shibata et al. [15] 2007	LUAD	*KRAS*	Bronchial
		EGFR	Alveolar
Angulo et al. [46] 2008	NSCLC	*PI3K3CA*	-
		*BRAF*	-
		*LKB1*	-
		*KRAS*	-
		*TP53*	-
		*EGFR*	EGFR mutated LUAD transcriptional cluster
Wilkerson et al. [20] 2012	LUAD	*EGFR*	Bronchioid
		*KRAS*	Magnoid
		*TP53*	Magnoid
		*STK11*	Magnoid
		*LRP1B*	-
		*BRAF*	-
		*PTEN*	-
Okayama et al. [48] 2012	LUAD	*EGFR*	-
		*KRAS*	-
		*ALK*	ALK positive cluster
Hammerman et al. * [19] 2012	LUSC	*KEAP1*	Classical
		*NFE2L2*	Classical
		*PTEN*	Classical, Primitive
		RB1	Primitive
Planck et al. [47] 2013	LUAD	*EGFR*	EGFR-1 and EGFR-2 subtypes
		*KRAS*	-
Collison et al. * [24] 2014	LUAD	*KRAS*	PP
		*STK11*	PP
		*NF1*	PI
		*TP53*	PI
		*EGFR*	TRU
Rignér et al. [25] 2016	LUAD	*EGFR*	CONSENSUS_1
		*KRAS*	CONSENSUS_2
Chen et al. * [26] 2017	NSCLC	*TP53*	SQ.2a, SQ.2b, AD.1, AD.2, AD.3
		*RASA1*	-
		*PTEN*	-
		*SMARCA4*	-
		*CDKN2A*	-
		*NFE2L2*	-
		*STK11*	AD.1, AD.5b
		*KRAS*	AD.2, AD.5b
		*KEAP1*	-
		*EGFR*	-
Hu et al. * [27] 2019	LUAD	TP53	Subtype 1, Subtype 2
		EGFR	Subtype 4

**Table 3 cancers-13-04734-t003:** EGFR targeted therapy resistance/sensitivity transcriptional signatures and proposed molecular mechanisms.

Study	Targeted Therapy	Target Gene	Sensitivity or Resistance	N Signature Genes	Main Proposed Mechanism
Coldren et al. [77] 2006	Gefitinib	EGFR	Both	Sensitivity: 305 Resistance: 105	E-cadherin upregulation in sensitive cell lines
Balko et al. [78] 2006	Erlotinib	EGFR	Sensitivity	180	Overexpression of MAPK and PI3K pathways
Zhang et al. [79] 2012	Erlotinib	EGFR	Resistance	21	*AXL* overexpression
Byers et al. [80] 2013	Erlotinib PI3K-i	EGFR PI3K	Resistance	76	*AXL* overexpression and epithelial–mesenchymal transition activation
Terai et al. [81] 2013	Gefitinib	EGFR	Resistance	-	FDF2-FGFR pathway activation
Geeleher et al. [82] 2014	Erlotinib	EGFR	Sensitivity	1000	Not highlighted
Liu et al. [83] 2015	Gefitinib	EGFR	Resistance	-	*IL-8* overexpression and enriched stemness properties
Rothenberg et al. [84] 2015	Erlotinib	EGFR	Resistance	35	*SOX2* overexpression
Naeini et al. [85] 2018	TKIs	EGFR	Resistance	1286	- Glycolysis and cell cycle upregulation - Immune response downregulation - Apoptosis downregulation - P53 pathway downregulation - TNF-alpha pathway downregulation - Xenobiotic metabolism downregulation
Cheng et al. [86] 2020	Erlotinib Gefitinib Other TKIs	EGFR	Sensitivity	11,431	Not highlighted

**Table 4 cancers-13-04734-t004:** Most relevant gene expression signatures associated with immune response prediction.

Study	Year	Description
Liberzon et al. [90]	2011	Immunologic signatures of MSigDB.
Bindea et al. [91]	2013	28 transcriptional signatures to quantify the degree of infiltration of different immune cell subpopulations. Both innate immune cells and adaptative immune cells are included.
Rooney et al. [92]	2015	CYT score based on the expression of the effector molecules that drive cytolysis.
Prat et al. [93]	2017	23 immune-related signatures linked to response and progression-free survival after treatment with anti-PD1 therapy.
Ayers et al. [94]	2017	IFN-gamma pathway gene signature related to antigen presentation, chemokine expression, cytotoxic activity, and adaptative immune resistance. It has been associated with clinical benefit upon anti-PD1 pembrolizumab treatment.
Thorsson et al. [95]	2018	Identification of 6 immune subtypes/signatures characterized by differences in tumor microenvironment and prognosis. Specific mutations correlated with lower (*CTNNB1*, *NRAS*, *IDH1*) or higher (*BRAF*, *TP53*, *CASP8*) lymphocytes presence.
Danaher et al. [96]	2018	Tumor Inflammation Signature (TIS) takes into account genes related to antigen presentation, cytotoxic activity, and adaptative immune resistance. The TIS has been shown to enrich for patients who respond to the anti-PD1 treatment pembrolizumab. The TIS has been applied retrospectively in multiple immuno-oncology trials.
Critescu et al. [97]	2018	T-Cell inflamed gene expression signature and TMB potential to jointly predict response to pembrolizumab in more than 300 patients with advanced solid tumors and melanoma from KEYNOTE trials.

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
