# Peer review of "Gene Expression Profiling as a Potential Tool for Precision Oncology in Non-Small Cell Lung Cancer"

_cancers, 2021, doi:10.3390/cancers13194734_

Round 1

Reviewer 1 Report

I have reviewed the changes made by the authors in response to my comments and the authors have adequately addressed my concerns.

Reviewer 2 Report

Authors improved the manuscript providing new data (both in main and suppl. materials). The presentation of the topic is satisfactory, therefore, I recommend for publication the revised version of manuscript presented by authors.

This manuscript is a resubmission of an earlier submission. The following is a list of the peer review reports and author responses from that submission.

Round 1

Reviewer 1 Report

The manuscript by Hijazo-Pechero et al. provides a comprehensive literature research, in attempt to demonstrate the utility of gene expression profiling as a potential tool for patient stratification and treatment guidance. The findings are intertesting and will be of interest to the wider readership of "cancers", and importantly provide ground for further research into personalising future trial design/therapeutic combinations in NSCLC patients.

Reviewer 2 Report

The review proposed by Sara Hijazo-Pechero et al. concerns the value of translational background in precision oncology of NSCLC. The topic is very interesting and a general presentation of the entire context is satisfactory. However, the description in too generic therefore I would suggest to remodel the text adding more detailed data and summarizing them in additional figures or tables. Moreover, I haveindicted several major and minor remarks to improve.  

Major remarks

  1. Figure 1 does not present the informative data only the overview that is more suitable for a supplementary material. More valuable would be a figure summarizing the description of paragraphs 2.1-2.3 and presenting how transcriptional subtypes are related with the druggable/driver alterations, biological processes, clinical outcomes or how the translational signatures overlap between LUAD and LUSC.
  2. Figure 2 should be presented in a better quality and need to be elaborated in details. In this form is not clear what the graph should present and how beneficial would be the evaluation of transcriptional signatures or what additional benefit they could add to the current way of patients qualification for personalized therapies. Please connect the translational signatures with druggable/driver alterations. It is also not clear what author mean as the tractable alterations. To make the figure more valuable authors should keep the track of published data.
  3. There should be also the list/table summarizing which translational signatures are related with currently available personalized drugs (both targeted and ICIs).
  4. Table 2 – instead of the articles’ title more beneficial would be specifying the genes included in the transcriptional pattern. This issue should be also explained in detail in the paragraph 3.1. ex EGFR mutations (or response to EGFR TKIs) is characterized by transcriptional pattern that include the following changes ….
  5. The paragraph 3.2 should be implemented by the description of the value of single cell approaches (single cell sequencing , spatial transcriptomic) in specifying the translational signatures.
  6. Paragraph 3.3 should deeply specified the correlations between transcriptional signatures and resistance to targeted therapies indicating specifically identifying which signatures are responsible for treatment failure.
  7. Paragraph 4 should be implemented by the point 4.2 with the additional description of relation between translational signatures and failure to immunotherapy (the same like in 3.3 was done for targeted therapies)

Minor remarks:

  1. Lanes 30 and 34 – Non-Small-Cell Lung Cancer does not need to be written in capital letters.
  2. Lane 65 – please double check if this way of the webpage citation is correct according to journal regulations
  3. Lane 68-69 – to be precise NSCLC is divided into two main pathological types: lung squamous cell carcinoma and non-squamous carcinoma, further divided into lung adenocarcinoma and large cell carcinoma.
  4. Lane 68-69 – according to TCGA nomenclature the abbrev for Lung adenocarcinoma and lung squamous carcinoma are LUAD and LUSC respectively. I suggest to use those abrevs. within entire manuscript.
  5. Lane 262-264: the 67% of patients with NSCLC present an oncogenic driver mutation who can benefit from targeted therapy is extremely high especially if we compare different races.

Reviewer 3 Report

This concise review is well written and nicely describes the state of research and the historical studies that have examined gene expression and genomic alterations in lung cancer. The authors suggest that whole transcriptome sequencing and development of deconvolution tools are necessary to analyze the composition of complex mixtures of cells for defining response to immune and other therapies, as well as mechanisms of resistance. It would also be appropriate and beneficial to include additional studies which utilize the approaches of single cell RNA sequencing and/or validation with multiplex IHC as being appropriate for this type of analyses. It is unlikely such analysis of complex cell mixtures can be fully understood without using these types of tools.

Minor: The simple summary uses "Non-Small Cell Lung Cancer" but the Abstract uses "NSCLC". Since the Introduction introduces Non-Small Cell Lung Cancer as NSCLC some consistency might be helpful.
